# Improvement of Salinity Tolerance in Water-Saving and Drought-Resistance Rice (WDR)

**DOI:** 10.3390/ijms24065444

**Published:** 2023-03-13

**Authors:** Yi Liu, Feiming Wang, Anning Zhang, Zhihao Chen, Xingxing Luo, Deyan Kong, Fenyun Zhang, Xinqiao Yu, Guolan Liu, Lijun Luo

**Affiliations:** 1Key Laboratory of Grain Crop Genetic Resources Evaluation and Utilization, Ministry of Agriculture and Rural Affairs, Shanghai Agrobiological Gene Center, Shanghai 201106, China; 2Shanghai Collaborative Innovation Center of Agri-Seeds, Shanghai Agrobiological Gene Center, Shanghai 201106, China

**Keywords:** salinity tolerance, drought, dominant male sterile, recurrent selection, water-saving and drought-resistance rice (WDR)

## Abstract

Rice is one of the most economically important staple food crops in the world. Soil salinization and drought seriously restrict sustainable rice production. Drought aggravates the degree of soil salinization, and, at the same time, increased soil salinity also inhibits water absorption, resulting in physiological drought stress. Salt tolerance in rice is a complex quantitative trait controlled by multiple genes. This review presents and discusses the recent research developments on salt stress impact on rice growth, rice salt tolerance mechanisms, the identification and selection of salt-tolerant rice resources, and strategies to improve rice salt tolerance. In recent years, the increased cultivation of water-saving and drought-resistance rice (WDR) has shown great application potential in alleviating the water resource crisis and ensuring food and ecological security. Here, we present an innovative germplasm selection strategy of salt-tolerant WDR, using a population that is developed by recurrent selection based on dominant genic male sterility. We aim to provide a reference for efficient genetic improvement and germplasm innovation of complex traits (drought and salt tolerance) that can be translated into breeding all economically important cereal crops.

## 1. Introduction

Soil salinization is an enormous global environmental challenge affecting agricultural sustainability [1]. More than 45 million hectares of irrigated land worldwide are affected by salinization (accounting for 20% of the irrigated land), further increasing every year [2]. The main factors leading to soil salinization are low and infrequent rainfall, increased evaporation, depletion of groundwater resources, and improper irrigation methods. The threat of salinization is more alarming in arid and semi-arid areas [3]. Due to climate change, soil salinization and drought will intensify. Salt stress has negative ecological and environmental consequences and may threaten food security in salt-affected areas to reduce crop yields.

Rice is one of the major staple food crops in the world. It is widely cultivated in Asia and is significant in ensuring global food security. In China, saline-alkali soils are widely distributed; they are of many types and cover large areas, about 100 million hectares in total. Such soils are an important land reserve resource, but their high salinity restricts the planting and growth of common crops [4]. With the continuous population growth, the available farmland area and freshwater resources are decreasing, and the environmental pressure on human survival will continue to increase. Under these circumstances, food crop shortage will become increasingly threatening. Therefore, breeding new rice varieties suitable for planting in saline-alkali land is an effective strategy to improve the utilization efficiency of saline-alkali land and to increase rice yield.

Rice is a salt-sensitive crop; thus, improving salt tolerance through genetic means has become one of the important research directions in rice genetics and breeding. Since rice salt tolerance is a quantitative trait controlled by multiple genes, the underlying genetic mechanism is very complex. Although our understanding of rice salt stress adaptation and its potential mechanisms has improved, only limited progress has been made in developing salt-tolerant rice varieties [5]. Based on previous studies, this paper reviews the recent developments on salt stress physiological and molecular mechanisms underlying its effects on rice growth, rice salt tolerance mechanisms, the identification and screening of salt-tolerant rice resources, and the research progress on breeding strategies for improving rice salt tolerance. Notably, similar to salt tolerance, the genetic basis of drought resistance is also extremely complex, involving multiple genes, higher-order regulatory networks, and epigenetic and post-translational modification mechanisms. To date, there has been little success in the successful breeding of drought-resistance rice varieties by molecular breeding methods [6]. However, selection under different target environments using the “lowland and upland rice hybrid breeding” strategy has generated several water-saving and drought-resistance rice (WDR) varieties that are widely cultivated in the affected areas [7]. This article also presents ongoing research on salt-tolerant WDR germplasm selection using a recurrent selection population carrying dominant genetic male sterility. We aim to provide a reference and a guide for the genetic improvement of rice complex traits toward increased salt tolerance and drought resistance.

## 2. Effect of Salt Stress on Rice Growth

In soils with high salinity, the soil osmotic pressure is often higher than in the intracellular pressure rice root hairs. This leads to difficulties in water absorption from rice roots, resulting in osmotic stress. A high concentration of Na^+^ will interfere with the absorption of other ions, such as K^+^, negatively affecting the ion homeostasis of rice cells and generating ionic stress [8]. When Na^+^ enters rice cells at high concentrations, complex secondary stresses on plants can be generated, including extensive reactive oxygen species (ROS) accumulation, destruction of proteins and nucleic acids, damage to the lipid membrane system, and metabolic disorders [8]. Studies have shown that salt stress inhibits the seed germination and seedling growth of rice. During rice growth and development, salt stress leads to root growth inhibition, premature leaf senescence, reduced plant height, fertility reduction, decreased chlorophyll content, and photosynthetic efficiency [9,10]. High salt concentration leads to a significant decline in rice yield or even a complete crop failure. In addition, rice grain quality deteriorates under high soil salt concentrations; its processing, cooking, and eating quality, and rice starch viscosity decrease significantly [11].

## 3. Salt Tolerance Mechanisms

Rice is a moderately salt-sensitive crop. However, to adapt to external high-salt stress, rice has developed various physiological mechanisms during evolution to resist salt stress and maintain its normal growth, mainly including osmolyte synthesis, the maintenance of ion homeostasis, the activation of antioxidant mechanisms, and stress signaling regulation [12]. 

### 3.1. Osmotic Adjustment

Soil water potential is significantly decreased by continuous salt accumulation. When soil salt concentration exceeds a critical value around the rhizosphere, a large water potential difference occurs, resulting in the difficulty of plant roots to absorb water. In addition, when the extracellular water potential is severely reduced, it leads to the outflow of intracellular water, resulting in cellular water loss [13]. To enhance water absorption and utilization, rice plants absorb inorganic ions from the environment or synthesize small organic molecules intracellularly to alleviate osmotic stress and maintain intracellular water balance. Under salt stress, plants can adjust osmotic pressure by absorbing inorganic ions from the rhizosphere, such as K^+^ and Ca^2+^, since sufficient K^+^ can quickly and effectively adjust the osmotic pressure of plants. The organic small molecule compounds synthesized in rice are mainly proline, betaine, and simple and complex carbohydrates (sugars) [14]. Under salt stress, the content of proline in rice is generally proportional to the resistance. 1-pyrroline-5-carboxylic acid synthase catalyzes the proline synthesis to improve rice’s salt tolerance [15]. Under normal conditions, the content of betaine in rice is very low. Under salt stress, betaine synthesis can be promoted through betaine dehydrogenase, which improves plant osmotic adjustment ability, protects the integrity of cell membrane, and promotes absorption of soil water by plant [16,17]. Soluble sugars are a group of important osmolytes. In rice, under salt stress, trehalose can be synthesized by trehalose-6-phosphate synthase, resulting in increased trehalose content and enhanced resistance to salt stress [18].

### 3.2. Ionic Homeostasis Mechanisms

Under salt stress, the high NaCl concentration in the soil causes excessive accumulation of Na^+^ inside the plants, negatively impacting the internal ionic balance and causing ion toxicity [19]. Under high salt conditions, plants can regulate the intracellular ionic balance by controlling Na^+^ absorption by the roots, mediating Na^+^ efflux, regulating Na^+^ distribution in different plant tissues/organs, or isolating Na^+^ to alleviate salt stress damage to plants [20]. The Na^+^ concentration increase in the aboveground plants is an important factor in the occurrence and severity of salt damage. When rice encounter high salt stress, the roots generally absorb more Na^+^ while activating salt resistance mechanisms. Several Na^+^ transporters and Na^+^/K^+^ exchange transporters can regulate the absorption of Na^+^ in the aboveground parts to maintain its concentration at a physiologically normal level, thereby improving rice resistance to salt stress [21]. When Na^+^ enters the aboveground tissues of rice, Na^+^ transporters unload Na^+^ in the vascular bundle for its long-distance basipetal transport. This process transports and compartments the excessive Na^+^ from the plant’s aboveground tissues to the root parenchyma, thus reducing the accumulation of Na^+^ in the photosynthetic tissues. Other mechanisms of this process include the isolation of Na^+^ in senescent leaves to reduce the damage to the young photosynthetically active leaves [22,23]. In addition, plants can promote Na^+^ efflux mediated by the SOS signaling and transport pathway, which is crucial for ionic balance. Rice contains two proteins that comprise the SOS pathway. SOS1 has the function of a Na^+^/H^+^ antiporter and regulates Na^+^ efflux. SOS2 encodes a serine/threonine protein kinase activated under salt stress. *OsCBL8* can interact with *OsCIPK24/OsSOS2* to form a protein kinase complex that regulates the *OsSOS1* antiporter located on the plasma membrane. As a result, excessive cellular Na^+^ is removed [24,25]. The membrane Na^+^/H^+^ transporter NHX transports Na^+^ ions and performs an important role in improving plant salt tolerance. H^+^ pyrophosphate and ATP proton pumps are localized in the rice vacuolar membrane. Under salt stress conditions, excess Na^+^ is pumped into the vacuole through Na^+^/H^+^ transporters, thus avoiding excessive Na^+^ toxicity in the cytoplasm [26]. *OsVP1* is a V-type ATPase located in the plasma membrane and helps export Na^+^ from the cytoplasm. Na^+^ accumulation in the leaves of *OsVP1*-overexpressing plants was reduced, thereby improving rice salt tolerance [27]. *OsNHX1* was rice’s first identified Na^+^/H^+^ antiporter gene; its expression was induced by salt stress treatment. It functions in Na^+^ and K^+^ transport from the cytoplasm into vacuoles to reduce their high concentration in the cytoplasm [28].

### 3.3. Resistance to Oxidative Stress

When rice is subjected to salt stress due to the increased toxicity and osmotic stress, a large amount of ROS is produced. The accumulation of harmful peroxides can severely damage plant cells [29]. To reduce or avoid excessive ROS damage, protective antioxidant enzymes, such as catalase (CAT), ascorbic acid peroxidase (APX), glutathione reductase (GR), and non-enzymatic antioxidant compounds, such as glutathione (GSH) and ascorbic acid (AsA), jointly are induced as a defense mechanism against oxidative stress in rice [30]. Under salt stress conditions, GR in plants catalyzes the formation of reduced glutathione (GSH), which can act as a reducing agent to repair the damage caused by salt stress [31]. APX genes and APX enzyme activity in rice are induced by salt stress. Consequently, the ability to remove hydrogen peroxide (H_2_O_2_) is enhanced. This protects seedlings from oxidative stress and performs an important role in the growth and development of rice under a high-salinity environment [32]. GDP-D-mannose pyrophosphorylase catalyzes the synthesis of AsA precursor GDP-D-mannose, leading to AsA synthesis in the leaves and enhancing salt tolerance.

### 3.4. Signal Molecules

Ca^2+^ is an important signal molecule for plant stress responses and performs an important role in salt stress adaptation. Under salt stress conditions, Ca^2+^ concentration in the cytoplasm increases, which initiates Ca^2+^ signaling pathways [33]. The *Arabidopsis* glucuronosyltransferase MOCA1 mediates the binding of glucuronic acid (GlcA) and inositol phosphoryl ceramide (IPC) to form sphingolipid GIPC outside of the plasma membrane. Under salt stress, extracellular Na^+^ ions combine with GIPC, which sequentially changes the cell membrane surface potential and activates the plasma membrane Ca^2+^ channels [34]. Ca^2+^ signaling molecules further activate the SOS3-SOS2-SOS1 pathway, which mediates Na^+^ efflux in the extracellular space and regulates plant adaptation to salt stress [33]. In recent years, several studies have further identified key components in the SOS pathway, critical for plant salt tolerance, including *AtANN4*, a Ca^2+^-permeable transporter [35]; a 14-3-3 protein [36]; VPS23A, a key component of the endosome sorting complex required for transport (ESCRTs) [37]; and BIN2, a glycogen synthetase kinase 3-like protein kinase [38]. These findings have deepened our understanding of the mechanism underlying plant salt tolerance regulation by Ca^2+^ signaling molecules. Under salt stress, *OsSOS3/OsCBL4* senses Ca^2+^ signals and, subsequently, phosphorylates *OsSOS2/OsCIPK24* to activate *OsSOS1*, which promotes Na^+^ efflux [39].

Plant hormones perform a crucial role in rice salt stress signal transduction pathways, especially ABA. Under salt stress, plants promote stomatal closure and reduce leaf and whole plant water loss through ABA− and H_2_O_2_− mediated stomatal regulation mechanisms [40]. At the same time, transcription factors containing ABA-response elements (ABRE) or ABRE binding protein/factors (AREB/ABF) are upregulated, thereby maintaining intracellular osmotic balance and enhancing rice salt tolerance [39].

## 4. Selection and Utilization of Salt-Tolerant Rice Germplasm Resources

Sri Lanka was one of the first countries to introduce and cultivate salt-tolerant rice germplasm. In 1939, ‘Pokkali’, the first salt-tolerant rice cultivar in the world, was cultivated with an ability to tolerance salt concentration more than 0.3% in the soil [41]. Since 1944, salt-tolerant rice breeding has been carried out in India, and several salt-tolerant rice varieties, such as ‘Kala Rata 1-24’, ‘Nona Bokra’, and ‘M114’, have been successively created [42]. Since 1970, the International Rice Research Institute (IRRI) has also committed to researching rice salt tolerance. From 9000 rice varieties and accessions, 10 salt-tolerant rice genotypes, such as ‘Pokkali’ and ‘Nona Bokra’, have been identified, and salt-tolerant rice varieties, including ‘IR46’, ‘IR4422-28-5’, and ‘CSR 23’, have been bred successively. Among them, ‘CSR 23’ can grow in a wide pH range, between 2–10, and has performed well in field trials in the Philippines and India [43]. These salt-tolerant rice germplasm resources are an important source material to mine for salt tolerance-promoting genes to further elucidate the molecular mechanisms underlying salt tolerance and breed modern rice varieties with salt tolerance. For example, *Saltol*, a salt-tolerant QTL from ‘Pokkali’, has been widely implemented for the genetic improvement of rice salt tolerance [44].

The identification and evaluation of rice germplasm resources with increased salt tolerance started in 1976 in China. In 1985, the Jiangsu Academy of Agricultural Sciences selected, from more than 500 materials provided by IRRI, a salt-tolerant variety, named ‘80-85’, then successively identified and bred 114 salt-tolerant varieties, such as ‘Zhuziqing’ and ‘Baiguzi’ [45]. From 2808 imported accessions, the Chinese Academy of Agricultural Sciences screened 103 salt-tolerant accessions, of which, ‘81-210’, ‘Lansheng’ and ‘American rice’ displayed excellent performance in the Jiangsu coastal area [46]. However, due to other non-desirable characteristics, such as poor agronomic traits, most salt-tolerant resources have not been appropriately exploited in breeding projects for salt-tolerant rice production and cultivation. In November 1986, a wild rice plant was found near the coastal area of Zhanjiang, Guangdong. After years of breeding and selection, a rice variety called ’Haidao 86,’ displaying excellent salt tolerance, was bred, and is able to grow in saline-alkali and infertile soils. This variety also exhibits flooding and waterlogging tolerance on marginal land, with an average yield reaching approximately 2250 kg/ha [47]. Therefore, it is considered a strategic germplasm resource for developing new salt-tolerant rice varieties.

## 5. Strategies for Salt Tolerance Improvement in Rice

### 5.1. Conventional Breeding

Conventional breeding strategies for salt-tolerant rice mainly include selecting offspring with satisfactory salt tolerance, and excellent yield and quality properties through crossing salt-tolerant germplasm with widely cultivated elite varieties. This process requires many years of self-pollination and backcrossing combined with the identification and selection of salt-tolerant phenotypes. Worldwide, scientists have successfully bred several salt-tolerant rice varieties with excellent yield and quality traits, which have been successfully introduced for cultivation. For example, IRRI has generated varieties including ‘CSR13’, ‘CSR10’, ‘CSR27’, ‘IR2151’, ‘Pobbeli’, ‘PSBRc 84’, ‘PSBRc 48’, ‘PSBRc 50’, ‘PSBRc 86’, ‘PSBRc 88’, and ‘NSIC 106’, which are currently cultivated in many countries [48]. Since the 1990s, breeders in China have also bred and introduced a number of salt-tolerant conventional rice varieties or rice hybrids, such as ‘Dongdao 4’, ‘Changbai 9’, ‘Changbai 10’, ‘Changbai 13’, ‘Liaoyan 2’, ‘Liaoyan 9’, ‘Liaoyan 12’, ‘Yanfeng 47’, ‘Jinyuan 85’, ‘Jinyuan 101’, ‘Yandao 12’, ‘Tengxi 138’, ‘Suijing 1’, ‘Suijing 5’, and ‘Jinnongda 30’ [49].

Mutation breeding is an effective method to develop rice varieties with improved salt tolerance and high yield as it does not encompass any ethical issues compared with transgenic technology [50]. Mutation breeding approaches include ionizing radiation, such as gamma rays, x-rays, neutrons, and chemical mutagens. A number of studies have shown that mutagenesis increases the genetic variability for salt tolerance in rice. Salt-tolerant rice varieties, such as ‘Shua 92’ [51] and ‘Basmati 370’ [52], have been developed using gamma irradiation. Takagi et al. identified a loss-of-function mutation associated with salt tolerance and developed the salt-tolerant variety ’Kaijin’ [53]. Compared with the reference salt-tolerant varieties ‘Pokkali’ and ‘Johna 349’, ‘NIAB-Roce-1’, and ‘PSR 1-84’, varieties generated by gamma radiation had higher yields under saline-alkali conditions [54]. Similarly, chemical mutagenesis has also been exploited and used to improve rice salt tolerance. The salt-tolerant rice mutant ‘*rst1*’, showing significantly high chlorophyll content and seedling biomass under salt stress, was identified from an ethyl methane sulfonate (EMS) mutant library [55].

Traditional breeding strategies have limitations, such as a long crossing and selection cycle, low selection efficiency, and poor predictability. Recently, with the developments in rice molecular genetics and genomics, especially functional genomics, an increasing number of QTLs and functional genes regulating rice salt tolerance have been identified, providing valuable gene resources for molecular genomics-assisted rice salt tolerance improvement [56].

### 5.2. Molecular Marker-Assisted Selection (MAS)

MAS can be implemented to accurately select target traits in the early breeding generations, thereby accelerating the breeding process. Multiple favorable genes can be pyramided using MAS to improve breeding efficiency. MAS can also significantly reduce the linkage drag that is commonly observed in the backcross breeding process and facilitate the effective introduction of single genes [57]. In the breeding process for the development of salt-tolerant, drought-tolerant, and disease-resistant cultivars, it is challenging to perform phenotypic assessments, as they may include destructive measurements leading to the death of some plants or seed failure. Thus, many individuals with excellent overall traits would be lost. MAS technology for stress tolerance breeding can achieve foreground selection for target QTLs or genes in early generations and delay the phenotypic identification of target traits. Therefore, this strategy is conducive to accumulating a large breeding population at the early breeding stages and accelerating the breeding process of fine varieties [58].

A total of 1011 salt tolerance related QTLs were identified in 52 QTL studies of rice. More than half of the salt tolerance QTLs were in the seedling stage. Salt tolerance QTLs at each growth stage were distributed on the 12 rice chromosomes. The phenotypic contribution rate of a single QTL ranged from 0.02% to 81.56%. A total of 167 QTLs had a contribution greater than 20%, and these occupied 22.0% of the total QTLs [59]. Gregorio and Senadhira used AFLP markers to analyze salt tolerance QTLs in a population of F_8_ recombinant inbred lines of the ‘Pokkali’/‘IR29’ cross [60]. A major QTL simultaneously controlling Na^+^ and K^+^ contents and the Na^+^/K^+^ ratio, named *Saltol*, was detected on rice chromosome 1. In many rice-growing countries, such as India, the Philippines, Thailand, Vietnam, Bangladesh, and Senegal, *Saltol* has been widely used in MAS breeding, mostly using marker-assisted backcrossing (MABC) technology. A brief description of the MABC process is as follows: the *Saltol* donor parent was crossed with the recipient parent, followed by three backcrosses. *Saltol* tightly linked markers were used for foreground selection in each backcross generation, and markers covering the rice genome were used for background selection. Finally, recombinant individuals with fixed *Saltol* donor alleles and enhanced salt tolerance were selected. In these MABC breeding practices, the phenotypic selection is often used in tandem to accelerate background recovery. The methods of individual QTL transfer, and simultaneous multiple QTL transfer, are also commonly used for QTL integration [61]. By using MABC technology, the *Saltol* locus has been introduced into popular elite varieties in different countries: *Saltol* was introduced into ‘BR11’ and ‘BRRI dhan28’ in Bangladesh [62], ‘AS996’ and ‘BT7’ in Vietnam [63], ‘Rassi’ in West Africa [64], and ‘Pusa Basmati 1121’ and ‘PB6’ in India [65,66]. In India, a project conducted by multiple agencies is currently ongoing to introduce *Saltol* into excellent high-yielding rice varieties [67]. Wu et al. used the BSA method to locate a major QTL (*qST1.1*) in the F_2_ population of the ‘Haidao 86’/’DJY1’ cross, which may significantly contribute to the salt tolerance of ‘Haidao 86’. The introgression lines with high salt tolerance obtained through MAS confirmed that the *qST1.1* region was associated with salt tolerance. A comparative analysis revealed that *qST1.1* is located within the *Saltol* locus genomic position, and it could not be ruled out that they are the same locus. Except for *Saltol*, most QTLs responsible for salt tolerance traits have relatively minor effects, which are greatly affected by the environment [68].

### 5.3. Genome Editing

In recent years, genome editing technologies via approaches, such as *CRISPR/Cas9,* have revolutionized genetic engineering, providing efficient tools for targeted and precise genetic improvement of crops. Since the *CRISPR/Cas9* technology is simple, flexible, and highly accurate, it has been effectively applied in key crops and model plants [69]. *CRISPR/Cas9* has been widely used in various crops. In rice, genome editing can be combined with innovation to create new varieties with improved yield, quality, and high resistance to abiotic and biological stresses. In terms of biotic and abiotic stress improvement, *CRISPR/Cas9* has shown great prospects in generating the required genetic modifications [70].

At present, more than 150 genes regulating rice salt tolerance have been identified and cloned [8]. Among them, at least 22 genes negatively regulate salt tolerance in rice. Compared with the wild type, EMS mutants, Tos17 insertion mutants, T-DNA insertion mutants, *CRISPR/Cas9*-editing mutants, or antisense RNA-knockdown plants and RNAi-knockdown plants targeting these genes all show enhanced salt tolerance. Thus, these genes are expected to be ideal targets for the molecular improvement of rice salt tolerance through genome editing. Based on the early findings of Huang et al. [71] and Takagi et al. [53] that *OsDST* and *OsRR22* negatively regulate rice salt tolerance, Santosh-Kumar et al. [72] created an *OsDST* 366 nt deletion mutant dst*Δ184-305*. Compared with the wild-type, the leaves of dst*Δ184-305* plants became wider, the stomatal density decreased, and the salt tolerance increased when plants were grown under 200 mmol/L NaCl. Zhang et al. [73] generated *OsRR22*-knockout mutant plants under a WDR genetic background. The salt tolerance of *osrr22* mutant plants was significantly higher than that of wild-type plants, both grown in a 0.75% NaCl solution. More importantly, under non-salt stress conditions, no significant difference in agronomic traits was found between *osrr22* and wild-type plants. Genome-edited DNA can be implemented to obtain mutant plants without transgenic components. With the rapid developments in functional genomics and the identification and characterization of more genes critical for stress tolerance, genome editing will provide more powerful and effective opportunities for improving crop salt tolerance.

## 6. Breeding of Salt-Tolerant WDR

For most plant breeders and physiologists, yield loss under stress is the most meaningful measurement of drought resistance or salt tolerance. Rice plants may adapt to drought or salinity through complex physiological or molecular mechanisms. These are very important for identifying specific traits related to drought resistance or salt tolerance and developing the appropriate screening techniques in breeding programs [74].

Drought resistance is a complex quantitative trait, including at least two mechanisms—drought avoidance and drought tolerance. Drought avoidance refers to plants’ ability to absorb or reduce water loss in water-deficient environments. This process requires water absorption through a sophisticated root system and water transport to the aboveground tissues of plants. Water consumption can also be reduced by moderately closed stomata or thickened cuticles. Drought tolerance refers to the ability of plants to maintain physiological and metabolic activities under water stress when leaf water potential decreases. Drought-tolerant plants actively accumulate osmolytes in the cells to increase their osmotic adjustment capacity and maintain high turgor pressure. In addition, they can also improve the ability to remove harmful compounds that are accumulating, and thus resist oxidative stress under drought conditions [75]. In fact, the drought resistance of plants is often a combined effect of the above properties. Drought avoidance performs a major role, and drought tolerance is considered the second line of defense for drought resistance [76].

Drought resistance is the result of a systemic network of interactions between multiple drought-resistance genes rather than the effect of a single drought-resistance gene. As mentioned in Pennisi [77], among numerous candidate drought-resistance genes revealed by genomics studies, almost none has significant impact on crop performance under field conditions, indicating that the research progress of molecular breeding on rice drought-resistance is slow. Therefore, selecting a breeding strategy that aggregates genes with different drought-resistance mechanisms is necessary. WDR is a newly cultivated rice type developed by introducing upland rice’s water-saving and drought-resistance characteristics based on technological progress in rice breeding science [75]. Wei et al. [78] also confirmed that drought-resistance genes and their networks could be transmitted in the offspring through conventional hybridization combined with high-intensity stress selections in the target environment. By using this strategy of crossing lowland rice and upland rice combined with selection under severe stress in the target environment, more than 20 new varieties of WDR have been generated. These WDR varieties are planted in dry and irrigated fields in China. Several varieties are successfully cultivated in Asian and African countries, and the planting area in China exceeds one million hectares [6,7]. The successful breeding of WDR provides a feasible solution for rice drought-resistance breeding.

The molecular and physiological mechanisms of plants’ responses to drought and salt stress exhibit similarities. Salt stress can cause ion and metabolic imbalance due to plant ion toxicity. Another salinity component is high osmotic stress, which leads to a water deficit similar to that caused by drought stress. Plants generally counteract the negative effects of salinity and drought by activating biochemical reactions, such as osmolytes synthesis and accumulation, intracellular ion homeostasis maintenance, and reactive oxygen species scavenging (Figure 1) [79,80]. Osmolytes performs important roles in plant response to abiotic stress. Enhancing the biosynthesis of trehalose can increase yield potential in marker-free transgenic rice under drought and salt conditions [81]. Heat shock proteins (Hsps) were also involved in osmotic adjustment. Overexpression of *OsHsp17.0* and *OsHsp23.7* enhances drought and salt tolerance in rice [82]. A novel C2H2 zinc finger transcription factor, *DST* (drought and salt tolerance), that controls stomatal aperture under drought and salt stress in rice. DST contributes to stomata movement via regulation of genes involved in ROS homeostasis [71]. *OsRLCK241* improved salt and drought tolerance in rice is mainly due to improved ROS detoxification, increased accumulation of osmolytes, and altered expression of stress-responsive genes [83].

Traditional cross or backcross breeding involves only a few parents are involved and the opportunities for excellent gene recombination are limited, resulting in a narrow genetic base of the developed cultivars [84]. As drought and salt tolerance are highly complex and quantitative traits that are governed by multiple genes/QTLs, and there are lots of cryptic beneficial alleles [85]. It is difficult to use and pyramid all these genes through limited cross breeding.

Here, we introduce a method for breeding salt-tolerant and WDR varieties using populations developed by recurrent selection based on dominant genic male sterility lines (Figure 2). Recurrent selection involving dozens of parents and multiple cycle of trait selection is an effective breeding approach to simultaneously broad genetic diversity and improve the quantitative traits. Recurrent selection is a systematic population improvement method. Through “selection-recombination and reselection-recombination” cycles, the frequency of favorable genes in the population gradually increases, improving the target traits in the population. The advantages of recurrent selection include rapid cycling and subsequent accumulation of favorable alleles from many parents into a single superior genotype and increased recombination and breaking of repulsion-phase linkages. Breeding practices in crops have shown that recurrent selection is the most effective method for population improvement and generation of germplasm with novel traits, especially for the simultaneous improvement of multiple complex agronomic traits [86].

The dominant genic male sterile WDR rice lines can be generated by crossing more than five rounds of backcrossing with the dominant genic male sterile lines (female parents) and the core WDR parents (male parents).F_1_ seeds are obtained by crossing the WDR genic male sterile lines (female parents) with genotypes, such as ‘Haidao 86’, ‘Dongxiang wild rice’, and salt-tolerant donors.F_1_ seeds are mixed and sowed, the rice plants are pollinated during the heading stage, and the seeds are harvested at maturity from the male sterile plants with decent agronomic traits to obtain the first-generation recurrent selection population.The first-generation recurrent selection population is planted in a dry land, relying only on rainfall throughout the growth period, without artificial irrigation. At the heading stage, artificial pollination is carried out. The seeds from the elite male sterile plants are harvested at the maturity stage to obtain the generation recurrent selection population.The second generation of the recurrent selection population is planted in saline-alkali land or coastal land, and the seeds from the elite male sterile plants are harvested to obtain the third generation of the recurrent selection population.According to the breeding objectives and selection pressure, different selection environments can be set to obtain recurrent selection populations, or fertile individual plants that meet the breeding objectives can be selected to conduct conventional breeding programs.

In the offspring, the genotypic frequency of dominant male sterile is always around half, so the performance of male sterile material is quite important and needs to be in the most adaptable background. First, we transferred the dominant genic male sterile gene into elite water-saving and drought-resistance rice varieties through multi-generation backcrossing, that are highly adaptable to the drought stress. Parental selection is critical, it is necessary to consider the complementarity of main agronomic traits, and the differences in ecology, region, and genetic relationship. Selection pressure is closely related to the genetic composition of the population. Selection pressure should be combined with the law of genetic variation of target traits to avoid the loss of excellent genotypes and maintain the level of genetic variation in the population. For quantitative traits controlled by multiple genes, such as drought and salt tolerance, the selection pressure should be appropriately reduced during early selection to fully recombine the micro-effect genes. However, the selection pressure should be gradually strengthened with the increase in the number of cycles of selection.

At maturity, the seeds from the male sterile plants were bulk harvested for the next round of recombination and seeds from male fertile plants were bulk harvested for screening under drought and salt stress conditions [87]. With this approach, we obtained a series of elite salt-tolerant and WDR lines, of which 16 exhibited Grade 1 salt tolerance at the budding stage, 10 exhibited Grade 1 salt tolerance at the seedling stage, and 14 exhibited extremely high or high salt-tolerance during the whole growth period (Table 1).

## 7. Conclusions and Prospects

Soil salt stress threatens rice production and global food security. Cultivating salt-tolerant rice varieties is the most effective approach to overcome this environmental hurdle. However, rice salt tolerance mechanisms are highly complex, involving various physiological and biochemical pathways. It is challenging to improve quantitative traits controlled by multiple genes through traditional breeding methods; thus, the current progress is slow. MAS and genetic engineering approaches can accelerate the process of breeding salt-tolerant rice varieties. However, most existing research has focused on transferring a single salt-tolerance QTL—*Saltol*, resulting in a simple genetic basis of salt tolerance in the bred varieties. Thus, it is difficult for these varieties to adapt to different types of saline land environments, which limits their large-scale cultivation. In addition, obtaining salt-tolerant varieties that can be used in field production by introducing a single gene or a few related genes is challenging.

Therefore, the effective generation of salt-tolerant rice varieties translated to increased field productivity may require the simultaneous introduction of multiple key genes, which allows the genetic improvement on multiple pathways of the salt-tolerance regulatory networks. This requires a breakthrough in breeding methods. The development of genome editing technologies makes it a routine operation to mutate multiple target genes simultaneously. Studies have shown that genes negatively regulating salt tolerance in rice involve different salt-tolerance pathways. Therefore, simultaneously knocking out multiple genes negatively regulating salt tolerance can integrate multiple salt-tolerance genes. This process is expected to stack the effects of different salt-tolerance pathways and achieve the generation of rice lines with stronger salt tolerance, suitable for cultivation in the field.

We further introduced the development of WDR by crossing lowland rice and upland rice combined with selection under severe stress conditions in the target environment. The successful development of WDR provides a feasible technical scheme and concept for the improvement of complex traits (drought resistance and salt tolerance) in rice cultivars. The breeding approaches towards the generation of salt-tolerant and WDR lines show that recurrent selection is an effective method for crop population improvement and the generation of lines with novel traits, especially regarding the simultaneous improvement of multiple complex agronomic traits.

## Figures and Tables

**Figure 1 ijms-24-05444-f001:**
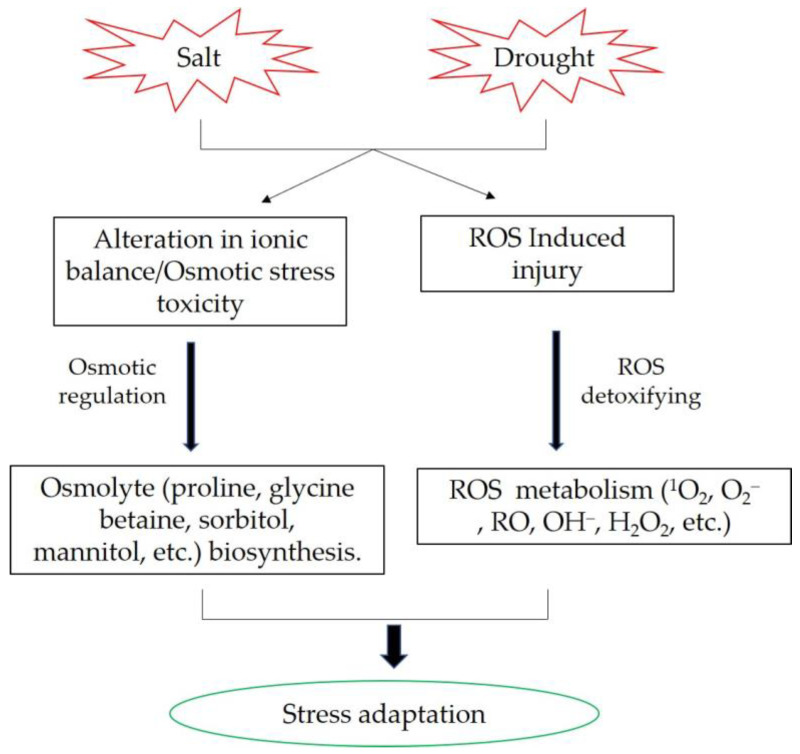
The common mechanisms involved in rice salt and drought stress responses.

**Figure 2 ijms-24-05444-f002:**
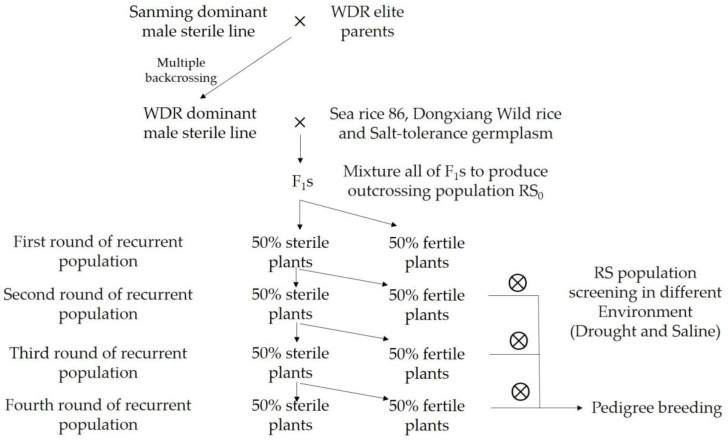
Improvement salt resistance of WDR by dominant male sterile recurrent selection population.

**Table 1 ijms-24-05444-t001:** Salt-tolerant WDR lines obtained by dominant male sterile recurrent selection system.

Lines	Plant Height (cm)		No. of Effective Spikes per Plant		Spike Length (cm)		Seeding Rate (%)		1000 Grain Weight (g)		Yield per Plant (g)		Salt Stress Index (%) ^a^	Salt Tolerance Score ^b^	Evaluation of Salt Tolerance ^c^
	Salt Treatment	Normal Control	Salt Treatment	Normal Control	Salt Treatment	Normal Control	Salt Treatment	Normal Control	Salt Treatment	Normal Control	Salt Treatment	Normal Control			
NYS5	74.0	81.0	12.0	14.3	17.3	18.7	68.1	82.6	20.8	22.7	12.9	15.6	17.2	1	HR
NYS6	93.0	104.6	14.6	18.6	15.7	17.3	60.1	80.7	21.5	24.7	8.6	12.3	30.1	2	R
NYS8	99.1	122.4	6.2	11.2	20.9	21.4	76.0	92.1	18.7	23.2	9.6	14.9	35.2	2	R
NYS10	83.2	96.8	10.0	13.8	17.9	19.1	63.6	85.0	18.9	23.0	6.7	8.6	22.1	2	R
NYS11	74.6	81.5	7.4	10.4	16.6	18.4	45.1	70.8	15.7	21.4	7.4	10.3	28.3	2	R
NYS15	79.1	86.3	4.4	7.0	19.0	19.8	63.8	79.4	18.6	20.5	13.5	18.5	26.9	2	R
NYS16	65.8	68.2	8.6	12.0	11.7	12.1	49.6	76.2	18.7	25.9	8.0	14.0	42.9	3	MR
NYS17	69.6	76.2	6.2	8.2	18.1	19.1	78.1	91.5	17.5	21.3	9.0	11.5	21.5	2	R
NYS23	67.1	78.3	10.0	14.8	12.3	13.7	58.5	77.4	16.3	22.1	8.3	20.3	59.1	3	MR
NYS39	84.8	102.7	7.2	8.6	19.1	21.0	82.5	95.8	18.0	19.6	12.0	17.5	31.3	2	R
NYS55	71.2	88.5	4.0	5.6	19.3	20.2	74.9	83.8	17.4	19.8	10.4	16.7	37.7	2	R
NYS73	77.7	95.1	7.8	9.8	19.2	20.6	73.7	79.1	18.2	22.7	11.2	25.5	56.2	3	MR
NYS83	72.4	84.7	10.8	13.8	16.7	18.1	65.6	83.6	19.6	22.8	18.6	23.3	20.4	2	R
NYS87	75.0	88.5	8.0	13.0	21.1	19.4	60.8	93.7	20.0	21.2	8.6	14.3	39.5	2	R
NYS92	81.0	99.3	8.0	10.8	20.4	21.4	80.6	93.7	19.0	21.6	12.6	16.7	24.7	2	R
NYS101	92.4	111.8	5.6	7.2	19.5	20.8	67.0	92.0	17.4	18.9	11.6	19.7	41.0	3	MR
NYS105	79.9	99.5	7.0	10.8	18.1	20.2	73.7	87.5	16.3	18.9	10.2	14.1	27.7	2	R
NYS117	67.6	79.6	7.2	9.2	18.0	19.4	71.5	87.0	17.1	20.8	8.5	10.3	18.2	1	HR
NYS120	77.6	108.7	8.8	12.2	18.4	21.5	68.9	84.3	19.1	22.1	7.2	14.6	50.4	3	MR
Huhan 61	62.3	68.6	9.3	9.4	12.4	12.7	42.9	79.8	18.2	25.2	6.8	16.2	58.2	3	MR
Huhan 3	60.3	73.3	6.8	9.4	12.2	12.6	48.5	81.0	18.7	25.2	7.8	13.6	42.6	3	MR
Sea Rice 86	93.0	105.7	16.0	20.3	15.3	17.1	65.7	77.4	22.1	24.8	12.7	15.6	18.8	1	HR

^a^: Salt stress index of yield per plant was used to determine the damage degree due to its susceptibility under salt stress. Salt stress index = (Yield under control condition − Yield under salt stress condition)/Yield under control condition × 100. ^b^: Salt tolerance score was classified based on salt stress index as Grade 1 (≤20.0%), Grade 2 (20.1–40.0%), Grade 3 (40.1–60.0%), Grade 4 (60.1–80.0%), and Grade 5 (≥80.1%). ^c^: HR, R, and MR mean high resistant, resistant, and medium resistant, respectively.

## Data Availability

All the data generated or analyzed during this study are included in this published article.

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
