# Peer review of "Improvement of Salinity Tolerance in Water-Saving and Drought-Resistance Rice (WDR)"

_ijms, 2023, doi:10.3390/ijms24065444_

Round 1
Reviewer 1 Report
I have some comments that maybe helpful for improving this manuscript before it can be accepted as below.
(1) P3, L108: It is hard to understand "promotes p soil water absorption".
(2) P3, L112; P4, L161; The subtitle "Ionic homeostasis mechanisms" was used twice.
(3) P3, L141; The sentence "resulting in improved rice salt " was incorrect.
(4) P3, L135, L142; The writing of "Na+/H+" was wrong. The authors should carefully review the manuscript and correct similar mistakes.
(5) It will be helpful to summarize the positioned salt-stress tolerance QTLs so far for readers to study and utilization.
(6) P8: In figure 1, the authors provided a diagram to show the common mechanisms involved in rice salt and drought stress responses. Does any genes were both involved in salt and drought stress response? If yes, please interpret its function mechanism and include in the manuscript.
(7) P9:In Figure 2, the arrows in 3rd and 4rd round of recurrent population were missed.
(8) P11: The Table 1 was hard to understand. A legend was necessary to introduce the how the Lines were obtained, how the salt stress tolerance index and salt tolerance score were acquired, what is the meaning of R, HR and MR.
(9) It seems that introduction of the dominant male sterility WDR lines just facilitated the crossing procedures except providing elite agricultural traits, and that the breeding period was not shortened because it would take multiple generations to obtain a stable line after crossing with salt-tolerant germplasms. So, more advantages of the method should be discussed and compared with traditional breeding method of crossing combination with backcrossing.
(10) Similar recurrent selection method has been used in dominant Taigu genic male sterile wheat, and there are application studies reported. Its application effect depends on the comprehensive agronomic traits of the dominant male sterile donor and the target gene donor and selection pressure. The final effect is not necessarily faster than that of directional backcross, and it is difficult to achieve the goal of polygene polymerization. It should be discussed by referring to the application effect on wheat.
Author Response
Response to Reviewer 1 Comments
Point 1: P3, L108: It is hard to understand "promotes p soil water absorption".
Response 1: We are very sorry for our careless mistake and it have been corrected. It means “promotes absorption of soil water by plant”.
Point 2: P3, L112; P4, L161; The subtitle "Ionic homeostasis mechanisms" was used twice.
Response 2: We thank the reviewer for pointing out this issue and it was rectified in P4, Line 162.
Point 3: P3, L141; The sentence "resulting in improved rice salt " was incorrect.
Response 3: Yes, the description of this sentence was incorrect. We have revised this sentence as: “thereby improving rice salt tolerance” in L142.
Point 4: P3, L135, L142; The writing of "Na+/H+" was wrong. The authors should carefully review the manuscript and correct similar mistakes.
Response 4: We apologize for the mistake of our manuscript. We have revised similar mistakes in the manuscript very carefully.
Point 5: It will be helpful to summarize the positioned salt-stress tolerance QTLs so far for readers to study and utilization.
Response 5: Thank you for this good suggestion. We have summarized this information in Line 267-271.
Point 6: P8: In figure 1, the authors provided a diagram to show the common mechanisms involved in rice salt and drought stress responses. Does any genes were both involved in salt and drought stress response? If yes, please interpret its function mechanism and include in the manuscript.
Response 6: Yes, there are some genes both involved in salt and drought stress response, such as osmoregulatory substance synthesis-related genes and reactive oxygen species-related genes. We have interpreted its function mechanism in the manuscript in Line 371-380.
Point 7: In Figure 2, the arrows in 3rd and 4rd round of recurrent population were missed.
Response 7: We thank the reviewer for pointing out this issue and we have added the arrows in Figure 2.
Point 8: P11: The Table 1 was hard to understand. A legend was necessary to introduce the how the Lines were obtained, how the salt stress tolerance index and salt tolerance score were acquired, what is the meaning of R, HR and MR.
Response 8: At maturity, the seeds from the male sterile plants were bulk harvested for the next round of recombination and seeds from male fertile plants were bulk harvested to perform phenotypic target trait screening. With this approach, we obtained 19 elite salt-tolerant and WDR lines, which exhibited good salt-tolerance during the whole growth period. The Salt stress index of yield per plant was used to determine the damage degree due to its susceptibility under salt stress. Salt stress index = (Yield under control condition − Yield under salt stress condition) ∕Yield under control condition × 100. Salt tolerance score was classified based on salt stress index as Grade 1 ( ≤ 20.0% ), Grade 2 ( 20.1% – 40.0% ), Grade 3 ( 40.1% –60.0% ), Grade 4 ( 60.1% – 80.0% ), and Grade 5 ( ≥ 80.1% ). HR, R and MR mean high resistant, resistant and medium resistant respectively.
Point 9: It seems that introduction of the dominant male sterility WDR lines just facilitated the crossing procedures except providing elite agricultural traits, and that the breeding period was not shortened because it would take multiple generations to obtain a stable line after crossing with salt-tolerant germplasms. So, more advantages of the method should be discussed and compared with traditional breeding method of crossing combination with backcrossing.
Response 9: As suggested by the reviewer, we discussed advantages of recurrent selection compared with traditional breeding method in revised manuscript in Line 387-402.
Traditional cross or backcross breeding involves only a few parents and the opportunities for excellent gene recombination are limited, resulting in a narrow genetic base of the developed cultivars. As drought and salinity tolerance are highly complex and quantitative traits that are governed by multiple genes/QTLs, and there are lots of cryptic beneficial alleles, it is difficult to use and pyramid all of these genes through limited cross breeding. Recurrent selection involving dozens of parents and multiple cycle of trait selection is an effective breeding approach to simultaneously broad genetic diversity and improve the quantitative traits. The advantages of recurrent selection include rapid cycling and subsequent accumulation of favorable alleles from many parents into a single superior genotype as well as increased recombination and breaking of repulsion-phase linkages.
Point 10: Similar recurrent selection method has been used in dominant Taigu genic male sterile wheat, and there are application studies reported. Its application effect depends on the comprehensive agronomic traits of the dominant male sterile donor and the target gene donor and selection pressure. The final effect is not necessarily faster than that of directional backcross, and it is difficult to achieve the goal of polygene polymerization. It should be discussed by referring to the application effect on wheat.
Response 10: Thanks for the reviewer's suggestion. we discussed the critical considerations for adopting recurrent selection based on dominant genic male sterility lines.
In the offspring, the genotypic frequency of dominant male sterile is always around half, so the performance of male sterile material is quite important and needs to be in the most adaptable background. We firstly transferred the dominant genic male sterile gene into elite water-saving and drought-resistance rice varieties through multi-generation backcrossing, that are highly adaptable to the drought stress.
Parental selection is critical, it is necessary to consider the complementarity of main agronomic traits, as well as the differences in ecology, region and genetic relationship. Selection pressure is closely related to the genetic composition of the population. Selection pressure should be combined with the law of genetic variation of target traits to avoid the loss of excellent genotypes and maintain the level of genetic variation in the population. For quantitative traits controlled by multiple genes, such as drought and salt tolerance, the selection pressure should be appropriately reduced during early selection to fully recombine the micro-effect genes. However, the selection pressure should be gradually strengthened with the increase of the number of cycles of selection.
Seeds from male fertile plants were harvested for screening under drought and salt stress conditions. The breeding period can be shortened by combining anther culture or rapid generation technology.

Reviewer 2 Report
My comments can be found in the attached MS. This manuscript provided more in-depth information on salinity tolerance and brief details were given on the draught tolerance aspect.

Author Response
Thanks for the reviewer comments. We have revised the manuscript carefully in line using the “Track Changes” function. Please see the attachment.
